# Asthma-COPD Overlap Syndrome: Recent Insights and Unanswered Questions

**DOI:** 10.3390/jpm12050708

**Published:** 2022-04-28

**Authors:** Evangelia Fouka, Andriana I. Papaioannou, Georgios Hillas, Paschalis Steiropoulos

**Affiliations:** 1Pulmonary Medicine Department, Aristotle University of Thessaloniki, G Papanikolaou Hospital, 57010 Thessaloniki, Greece; 22nd Department of Respiratory Medicine, “Attikon” Hospital, Medical School, National and Kapodistrian University of Athens, 12462 Athens, Greece; papaioannouandriana@gmail.com; 35th Pulmonary Department, “Sotiria” Chest Diseases Hospital, 11527 Athens, Greece; ghillas70@yahoo.gr; 4Department of Respiratory Medicine, Medical School, University General Hospital Dragana, Democritus University of Thrace, 68100 Alexandroupolis, Greece; steiropoulos@yahoo.com

**Keywords:** asthma, COPD, ACO, biomarkers, treatable traits, definition, pathophysiology, genetics, prognosis

## Abstract

The term asthma-COPD overlap (ACO) has been used to identify a heterogeneous condition in which patients present with airflow limitation that is not completely reversible and clinical and inflammatory features of both asthma and chronic obstructive pulmonary disease (COPD). ACO diagnosis may be difficult in clinical practice, while controversy still exists regarding its definition, pathophysiology, and impact. Patients with ACO experience a greater disease burden compared to patients with asthma or COPD alone, but in contrast they show better response to inhaled corticosteroid treatment than other COPD phenotypes. Current management recommendations focus on defining specific and measurable treatable clinical traits, according to disease phenotypes and underlying biological mechanisms for every single patient. In this publication, we review the current knowledge on definition, pathophysiology, clinical characteristics, and management options of ACO.

## 1. Introduction

Although Chronic Obstructive Pulmonary Disease (COPD) and asthma are two different entities, they often coexist in a single individual. Thus, in some COPD patients, predominant type 2 (T2) inflammation, in combination with significant reversibility and increased eosinophils in both blood and airways, can be recognized [1]. Similarly, other patients with a diagnosis of asthma and significant exposure to cigarette smoke may present some characteristics of COPD, such as fixed airflow obstruction, reduction in the diffusing capacity of the lungs for carbon monoxide (DLCO), and emphysematous lesions on chest CT [2]. The term asthma-COPD overlap (ACO) was first introduced by Gibson et al. [3] in 2009 to describe these patients, and since then this concept has gathered much interest, with more than 870 publications on the subject (PubMed search as of 15 April 2022).

ACO is a heterogeneous condition describing patients with persistent airflow limitation and clinical and inflammatory features of both asthma and COPD. Importantly, since significant smoking (or other exposure to noxious particles and gases) is essential for the diagnosis of COPD, it is also a precondition for ACO. This condition is necessary for not including in the diagnosis of ACO patients with asthma and non-fully reversible airflow limitation [4,5,6]. However, despite the increased research interest in ACO, there is still much controversy about its existence as a distinct entity, and from 2020 the Global Initiative for Chronic Obstructive Lung Disease (GOLD) stopped referring to ACO, emphasizing the absence of a universally accepted definition due to the lack of substantial evidence [7].

Nevertheless, despite the complexities regarding ACO definition, the diagnosis of this condition remains extremely important, since these patients seem to have a worse clinical outcome and experience a greater health impairment compared to patients with asthma or COPD alone [8]. Furthermore, the optimal management of the disease is mostly unknown, because, traditionally, patients with ACO have been excluded from most clinical studies [9]. In the present review we aim to summarize the current knowledge about ACO diagnosis, clinical characteristics, and current therapeutic interventions.

## 2. Definition

A number of authors and different consensus guidelines have proposed various definitions of ACO during the last decades, usually using several major and minor criteria [3,10,11,12,13,14,15,16]. Most of them emphasize the heterogeneity of clinical features and functional and laboratory findings that should be considered as diagnostic criteria that could apply for the identification of individuals with a possible diagnosis of ACO.

The main criteria in most of these reports include non-fully reversible airflow limitation, a history of asthma diagnosed before the age of 40 years, persistent symptoms that vary and progress over time, smoking history ≥ 10 years or significant exposure in other noxious particles or gases, high levels of eosinophils in sputum and/or in blood, exhaled breath nitric oxide (FeNO) levels ≥ 45 to 50 ppm, history of atopy or high IgE for total serum or inhaled allergens, metacholine challenge test positivity, and significant bronchodilation response [17].

Table 1 summarizes the proposed diagnostic features of ACO in the literature.

In the majority of ACO definitions, it is recognized that these individuals are suffering from smoking related COPD, with underlined T2 inflammation usually expressed with increased numbers of blood or sputum eosinophils [1,18,19]. However, other patients with a history of asthma and a significant smoking history may develop specific characteristics of COPD, such as airflow limitation, which is not fully reversible [4,5].

The most widely accepted ACO definition is the one of the 2015 GINA/GOLD documents. In this report, the patient should have a chronic airway disease (medical history of cough, sputum production, wheezing, or repeated lower respiratory tract infections) in combination with features of both asthma and COPD [20]. However, in the 2021 GOLD report the term ACO no longer exists and the authors state that COPD and asthma are two distinct entities which can coexist in a single individual [7]. Despite the removal of ACO from the most important reports on the management of asthma and COPD, i.e., the Global Initiative for Asthma (GINA) and the GOLD, this entity still continues to be important and receive attention in both clinical research and clinical practice.

In the general population, the prevalence of ACO ranges from 0.9% to 11.1%. Among asthmatics, ACO can be recognized in 11.1–61.0% and among COPD patients from 4.2–66% [14,19,21,22]. The main reason for such big differences between studies is the fact that each study uses a different definition for ACO [5].

## 3. ACO Pathogenesis

### 3.1. The Overlapping Mechanisms Hypothesis

The association between asthma and COPD was first stated in 1961 by the Dutch Professor Dick Orie, who proposed that “asthma, chronic bronchitis, and pulmonary emphysema is a single disease that occurs as a result of the same genetic factors (atopic status, promotion of airway hyperreactivity), and only presents different clinical phenotypes due to different environmental factors (allergens, smoking, and infections)” [23]. This became known as the “Dutch Hypothesis”, suggesting that obstructive airway diseases share a common genetic pathway, with bronchial hyperresponsiveness being important in both asthma and COPD, while environmental factors determine whether one would have asthma or COPD. This could be differentiated from the “British Hypothesis” provided in 1965 by Charles Fletcher [24], who proposed that an individual with certain genetic predisposition may present the clinical phenotypes of COPD and asthma after exposure to smoking and antigen sensitization, respectively.

### 3.2. Risk Factors for ACO

As discussed previously, the etiology of ACO likely involves the interactions between various host and environmental factors, beginning as early as fetal development and continuing into mature age [25]. Results from birth cohorts and cross-sectional studies have shown that airway diseases can occur during early life and even in utero, as a result of insults including maternal smoking and poor nutrition; subsequent acute viral infections and exposure to perennial aeroallergens can lead to persistent airway obstruction up to mid-adulthood in susceptible subjects [26,27]. Likewise, ageing can contribute to steroid-resistant airflow obstruction, related to increased lung compliance, and reduced lung elastic recoil, in predisposed non-smoking asthmatics [28,29], while some patients with asthma may develop an irreversible airflow obstruction, possibly through progressive airway structural changes and remodeling, after exposure to cigarette smoke or air pollution [30,31,32,33,34].

Smoking is strongly associated with ACO pathogenesis and a smoking history of ≥10 pack-years is one of the several newly proposed criteria for ACO [35]. It is well established that smoking is a risk factor for both asthma and COPD, by reducing treatment response to ICS and accelerating lung function decline [36,37,38]. Furthermore, smoking asthmatics present more symptoms compared to those who never smoked [39] and bronchial hyperactivity in continuing smokers with COPD is associated with a faster lung function decline [40].

Conclusively, evidence so far has not specifically addressed the unique pathophysiologic mechanisms responsible for persistent expiratory airflow limitation in patients with ACO. It is yet to be determined whether this entity develops gradually as the result of chronic airway inflammation and remodeling in a patient with COPD, as the consequence of noxious exposures in a patient with asthma, or as a de novo disease with its own pathology.

### 3.3. ACO Pathology

While asthma and COPD appear as a result of different mechanisms triggered by different pathogeneses and they present different features and symptoms of airway inflammation and airway obstruction, ACO is identified in clinical practice by features it shares with both asthma and COPD, that is, persistent airway limitation and increased reversibility [30]. The scarcity of data indicating discrete structural and immunological features in patients with ACO reflects the lack of evidence regarding ACO pathogenesis. The issue is not just whether pathologies typical of asthma or COPD co-exist in the same person, but also if such a common set of pathological characteristics contribute to the impaired physiology, namely COPD with increased reversibility or asthma with fixed airway obstruction.

An alternative approach to establish ACO existence would be to look for differences in biomarkers of tissue inflammation between ACO and asthma or COPD patients. In COPD, inflammation is predominately mediated by T-helper 1 (Th1) cells, while in asthma T-helper 2 (Th2) and type 2 innate lymphoid (IL2) cells are the key players of the inflammatory response [41].

In an attempt to address a potential histological characterization of ACO, a group from Basel performed biopsy procedures in a small cohort of COPD patients and demonstrated that specific histopathological changes, such as thickening of reticular basement membrane, may reveal a distinct ACO phenotype with higher response to ICS/LABA [42]. In this study, the ACO patients also demonstrated more elevated blood eosinophil counts and FeNO concentrations and higher reversibility and DLCO% predicted, as compared to COPD patients without asthma characteristics, suggesting a possible different underlying T2 pathophysiology.

According to a Spanish study, greater airway responsiveness and higher peripheral blood eosinophil counts and serum IgE levels were found in COPD patients with a prior diagnosis of asthma [43]. Moreover, Li et al. have found that among 48 patients, more than half of whom had a history of smoking and ICS use, FeNO levels > 31.5 ppb could differentiate patients with ACO from patients with COPD alone, with a sensitivity of 70% and a specificity of 90% [44]. Similarly, Shirai et al. have demonstrated that both high serum periostin and YKL-40 levels were significantly higher in ACO than in asthma or COPD patients [45]. However, in the largest comparative cohort study to date, NOVELTY, no difference in blood eosinophil counts was found between the asthma, COPD and ACO patients [46], suggesting that this biomarker is not useful in routine clinical practice in order to discriminate patients with suspected ACO. To date, there are no biomarkers that can clearly differentiate ACO from asthma or COPD.

Although, until recently, emphysema was assumed to be a characteristic feature of COPD that rarely appeared in asthmatic patients [47,48], small airways’ involvement in ACO is now evident by several studies that evaluated airway biopsy specimens, as well as findings acquired by novel imaging and non-invasive techniques, for a more effective assessment of small airways dysfunction in patients with asthma and COPD overlap [49]. Hartley et al., in their single-center study of 171 asthmatics, 81 COPD patients, and 49 healthy subjects with the use of quantitative computed tomography, showed that the strongest predictor of lung function impairment in asthmatic patients was the increase in airway wall thickness, while the degree of air trapping was the main driver of FEV_1_ decline in patients with COPD, suggesting that different pathological changes may contribute to the underlying physiology [50]. In a similar study, a Japanese group evaluated 167 patients with COPD, among whom 43 fulfilled the criteria for ACO, with the use of impulse oscillometry and three-dimensional CT imaging [51]. In this study, patients with ACO presented with higher respiratory resistance and reactance, greater morphological airway changes in central bronchi, and fewer emphysematous changes than COPD patients, even if they had a comparable smoking history and fixed airflow limitation, a finding suggesting a possible distinguished pathogenetic mechanism.

### 3.4. Genetics

Many genetic variants associated with risk factors for both asthma and COPD (ADAM33, TNFα, MMP9, TGFβ1, GSTM1, GSTP1) have been identified through genome-wide association studies (GWAS) [52,53]. A recent case-control study has also suggested that SNPs from the ADAM33 (V4 [rs2787094], T1[rs2280091], S2[rs528557]) and AQP5 genes (rs3736309) were specific markers for asthma and COPD, respectively, that may be used to predict individual predisposition [54].

Although there is still uncertainty whether ACO is a distinct clinical entity with discrete underlying genetic factors or an intermediate syndrome with overlapping genetic determinants within the asthma and COPD spectrum, emerging data contribute to the concept of the existence of a specific genetic architecture that, under the influence of certain environmental triggers, may be accountable for this phenotype.

The strongest evidence to date comes from the COPDGene study, a large GWAS that investigated potential characteristic genetic features of ACO in non-Hispanic whites and African-American populations [55]. Of the 3570 COPD subjects with GOLD severity ≥ 2, approximately 13% (*N* = 450) had ACO, as defined by a physician diagnosis. The authors identified SNPs strongly related to ACO in the genes CSMD1 (rs11779254,) and SOX5 (rs59569785) in the non-Hispanic whites and the SNP rs2686829 on chromosome 7 in the African-American population, while in the prespecified meta-analysis SNPs associated with ACO were identified in the gene GPR65 (rs6574978). Hansel et al. [56] performed another GWAS in European Americans with COPD from the Lung Health Study (LHS), intending to detect genetic markers of airway hyperresponsiveness, a cardinal feature of ACO. Immunohistochemistry identified risk alleles on chromosomes 5 and 3 that acted as expression quantitative trait loci (eQTLs) for SGCD and MYH15 messenger RNA proteins, relevant to the development of airway responsiveness and expressed in airway epithelium, vascular endothelium, and inflammatory lung cells. However, none of the SNPs found in this study reached the genetic threshold for statistical significance. Similarly, responsiveness to ICS, another characteristic feature of ACO, was examined in Lung Health Study-2 [57], a randomized controlled trial that evaluated the genetic determinants for lung function decline over 3 years, in which approximately 199 COPD patients randomized to fluticasone or placebo. Obeidat et al., showed that the SNP rs111720447 was significantly associated with rate of FEV1 decline both in patients taking ICS and in patients assigned to placebo; however, their findings also did not reach genome-wide significance. Finally, in a large GWAS of 35,735 cases and 222,076 controls, Sakornsakolpat et al. [58] identified genetic segments shared by asthma and COPD near ADAM19, ARMC2, ELAVL2, and STAT6 genes. On the contrary, in a GWAS of 15,256 cases and 47,936 controls, Hobbs et al. [59] did not identify significant loci for COPD overlapped with known loci for asthma, in European ancestry subjects. The discrepancy observed between these studies may be attributed to the difference in sample sizes between asthma and COPD groups or to the lack of assessing many environmental confounders that may explain, at least to some extent, the different pathogenesis of both diseases.

### 3.5. Clinical Manifestations—Diagnosis—Differential Diagnosis

According to the Global Initiative for Asthma (GINA), the evaluation of possible ACO resembles the evaluation of asthma and COPD [7]. The steps that need to be followed are summarized below:

### 3.6. Medical History

A medical history should include at first characteristics like the severity, frequency, and duration of respiratory symptoms and/or exercise limitation. Any prior diagnosis or symptoms of asthma or allergic rhinitis should be explored, along with exposures to fumes or dusts (like tobacco smoke, occupational exposures etc.)

### 3.7. Laboratory Tests

Blood examinations: Elevated levels of serum immunoglobulin-E (i.e., IgE >100 IU/mL), or peripheral blood eosinophil count (>300 cells/mL), accompanied by evidence of allergic disease (e.g., skin testing or immunoassays for perennial allergen sensitivity) may indicate an asthma or ACO diagnosis. Moreover, elevated sputum eosinophil counts are more common in asthma or ACO than COPD. In case of fixed airflow limitation, alpha-1 antitrypsin levels examination is suggested.

Pulmonary function tests (PFTs): PFTs before and after bronchodilation is an essential component of the diagnosis, in order to confirm airflow limitation (namely FEV_1_/FVC < 0.7). However, airflow limitation cannot help differentiating between ACO, asthma, and COPD. In some cases, after bronchodilation, the FEV_1_/FVC ratio may be ≥0.7 in ACO patients; however, this is more frequently seen in asthma. An increase in FEV_1_ ≥ 12% and ≥200 mL after bronchodilation is a common finding in ACO, especially if the initial FEV1 is low. However, an increase in FEV1 > 400 mL is indicative of asthma rather than ACO. Usually, in ACO patients post bronchodilation FEV1 is <80% predicted, but this is not a specific finding. In patients with ACO a greater increase in pulmonary function with inhaled glucocorticoid (ICS) is usually observed, compared with COPD patients. Reduced DLCO is more commonly seen in COPD than in asthma, but so far, no specific cut-off point has been defined for ACO.

Imaging tests: The chest radiograph may show hyperinflation in some cases of ACO patients; however, it is not useful in the differential diagnosis among asthma, COPD, and ACO. Chest high-resolution computed tomography (HRCT) may be used in case of a diagnostic uncertainty. Small airway disease, in the absence of emphysema, could lead to a diagnosis of ACO, although other diagnoses, like asthma or bronchiolitis obliterans should be considered. Severe emphysema is not consistent with ACO diagnosis.

### 3.8. Diagnosis

Although ACO refers to the combination of asthma and COPD, no specific analogy of features that confirms ACO diagnosis has been set [9]. Despite the lack of a fixed definition, the following features are indicative of ACO diagnosis [10]:Age ≥ 40 years.Persistent respiratory symptoms (chronic cough, sputum, dyspnea, wheezing). These symptoms may vary.Airflow limitation not fully reversible, but with historical variability: Postbronchodilator FEV1/FVC < 0.7 or lower limit of normal and bronchodilator increase in FEV1 > 12% and 400 mL.Past history of doctor-diagnosed asthma.History of atopy or allergies.Exposure to a risk factor (e.g., tobacco smoking ≥10 pack-years or equivalent indoor/outdoor air pollution).

In Figure 1 is presented a schematic approach for investigating and diagnosing asthma and COPD overlap.

#### Differential Diagnosis

The differential diagnosis of ACO includes other airway diseases such as bronchiectasis, obliterative bronchiolitis, central airway obstruction, and diffuse panbronchiolitis [10].

Bronchiectasis is suspected on the basis of symptoms like chronic productive cough, mucopurulent sputum, recurrent chest infections, and, less frequently, hemoptysis. The diagnosis is usually established based on characteristic findings of bronchial wall thickening and luminal dilatation seen on chest HRCT scans.Bronchiolitis obliterans is characterized by concentric fibrotic narrowing of the bronchiolar lumen. It is commonly seen after a viral illness, an inhalation injury, transplantation (eg., bone marrow, lung), or in the context of rheumatic disease. Symptoms include progressive onset of cough and dyspnea, and hypoxemia at rest or with exercise. Findings on chest HRCT scans can include centrilobular bronchial wall thickening, bronchiolar dilation, tree-in-bud nodularity, and a mosaic pattern of attenuation of lung tissue density.Central airway obstruction is attributed to a number of benign and malignant processes with slowly progressive dyspnea on exertion, followed by dyspnea under minimal activity. A flow-volume loop, which can be insensitive, and CT with 3-dimensional reconstruction may be helpful, but the gold standard for diagnosis is direct visualization.Diffuse panbronchiolitis is characterized by bronchiolitis and chronic sinusitis and it is highly prevalent in Asiatic populations. A prominent clinical feature is cough productive of copious amounts of sputum.

### 3.9. Challenges in ACO Diagnosis

As not a single factor as a respiratory symptom, specific biomarker or spirometric parameters can differentiate between asthma, COPD, and ACO, the diagnosis of the latter may be challenging, especially in patients with asthma who have developed irreversible airflow limitation, in patients with COPD with a history of asthma, in the elderly and in current or former smokers [60]. It could be argued that it may not be important whether there is a confident ACO diagnosis, as long as the clinicians can identify a group of patients who should be managed differently; however, this approach runs the risk of making the facts fit the prejudice of the observer, as in such a case ACO diagnosis is axiomatic. On the other hand, a very strict bronchodilator reversibility criterion excludes so many patients, that the definition has to be relaxed to allow the identification of anyone with ACO [61].

A different approach was attempted by Pascoe et al. [62], who investigated the ability of a statistical model approach to define distinct disease groups in patients with obstructive lung disease, with the use of medical histories and spirometric data. The resulting model was accurate in distinguishing asthma and COPD, but the authors suggested that patients not falling into these groups were very heterogeneous and hard to be classified. A similar substantial heterogeneity was emphasized by the results of the NOVELTY study [46], which showed that in the ‘real world’ diagnostic groupings are not rigidly applied.

Physicians are more likely to label asthma with fixed airway obstruction as COPD than to label COPD with partially reversible airway obstruction as asthma [63]. Therefore, a proportion of patients with a pre-existing diagnosis of COPD may have indeed asthma or ACO; it is necessary to identify this subgroup of patients because they have specific treatment needs [61]. However, caution should be taken when interpreting reversibility testing results in patients with COPD, because the prevalence of positive reversibility testing varies according to the criteria used and some COPD patients may also exhibit significant reversibility of lung function following the administration of short-acting bronchodilators [64].

On the other hand, ACO should be considered in patients with asthma and risk factors for developing persistent airflow limitation: childhood asthma with persistent wheeze from the first years of life, long-standing asthma without taking controller therapy with ICS, adult-onset asthma; and severe or difficult-to-treat asthma [65]. Nevertheless, a history of self-reported, primary care physician-diagnosed asthma is not reliable in patients aged ≥40 years because it is subject to recall bias and because not all patients with asthma have typical symptoms that respond to ICS, neither have had a definitive diagnosis [66].

Finally, diagnosis or severity evaluation is often delayed in elderly patients with ACO because, although spirometry can be adequately performed in more than 90% of older patients with obstructive airway disease, it is still hampered by poor cooperation in this population [67].

## 4. Treatment—Prognosis

The treatment of patients suffering from ACO has always been associated with the origin of the co-existence of the two diseases: which disease’s initial mechanisms eventually lead to ACO?

The Science Committees of both GINA and GOLD, based on a detailed review of available literature and consensus developed a document for ACOS (Asthma COPD Overlap Syndrome) in 2014 [10]. This document presented features that identify and characterize ACOS, ascribing equal weight to features of asthma and of COPD. A simple approach to initial treatment of ACOS was also included. Based on these recommendations initial treatment should be selected to ensure that (a) patients with features with asthma receive adequate controller therapy including inhaled corticosteroids (ICS) but not long-acting bronchodilators as monotherapy and (b) patients with COPD receive appropriate symptomatic treatment with bronchodilators or combination therapy, but not ICS as monotherapy.

In 2017, GINA formally recommended use of the term “ACO” rather than “ACOS”, to avoid giving the impression that this is a single disease or a syndrome. GINA recommendations 2021 for patients with features of both asthma and COPD suggest treatment as asthma: ICS-containing therapy is essential to control the disease and reduce the risk of severe exacerbations and death, LABAs (long-acting bronchodilators) and/or LAMAs (long-acting muscarinic antagonists) should not be used alone without ICS [7].

Patients with ACO have both asthma and COPD characteristics, and distinguishing asthma from COPD can be difficult especially in older smokers. Thus, the treatment of this special group of patients has to be counted on the crown stones of each disease separately: (a) long-acting bronchodilators are the central part of COPD treatment and only a small percentage of COPD patients will benefit from single long-acting bronchodilators, named anticholinergics. (b) The grand majority will continue to experience symptoms and/or exacerbations and for these patients the maximal (otherwise dual) bronchodilation is the appropriate choice. The maximal bronchodilation is a certain choice for ACO patients based on COPD side [68]. On the other hand, anti-inflammatory treatment with ICS is the central part for asthma patients. ICS are essential either alone or in combination with LABAs and/or LAMAs to reduce the risk of severe exacerbations and death. Treatment with ICS is a certain choice for ACO patients based on asthma side [68]. Despite maximal inhaled treatment (triple therapy), some patients with co-existing asthma and COPD may still exacerbate. An option could be the addition of macrolides, especially in the neutrophilic endotype of “ACO” subgroup [69].

Based on the above, triple therapy with ICS/LABA/LAMA seems to be the appropriate inhaled choice for the treatment of patients with ACO. All ACO patients should be provided with detailed and structured education focusing on adherence and inhaler technique, which should be assessed before concluding that the current therapy is insufficient.

Patients with ACO with persistent symptoms and/or exacerbations despite triple therapy should be evaluated for features, such as sensitivity to perennial allergens, elevated total serum IgE, and/or peripheral blood eosinophilia, that might suggest a response to one or more of the biologic agents that have been developed for severe uncontrolled asthma. There are limited data on the use of biologic agents in patients with features of ACO, restricted to omalizumab [70,71,72,73,74]. Mepolizumab, another biologic therapy that blocks IL-5, has also been used in patients with eosinophilic COPD [75] with greater reduction in exacerbation rate observed in patients with higher levels of eosinophils at screening. In the pre-specified meta-analysis of individual patient data from METREX and METREO studies, blood eosinophil counts ≥150 cells/µL at screening or ≥300 cells/µL in the prior year allow for the identification of patients who are likely to benefit from mepolizumab therapy [76]. In contrast, the use of benralizumab, a biologic directed against the alpha-subunit of the IL-5 receptor, did not improve exacerbation rates in patients with moderate to very severe COPD and frequent moderate or severe exacerbations, regardless of eosinophil level [77]. Based on these findings, the usefulness of biologics in the management of ACO remains uncertain.

The management of ACO patients includes non-pharmacological interventions as well: physical activity, pulmonary rehabilitation (especially in patients with features likely to be COPD), vaccinations (flu, pneumococcal, COVID-19), smoking cessation, diet modification, and guideline-based management for specific comorbidities.

As patients with ACO often have worse outcomes (more frequent or severe exacerbations, accelerated loss of lung function, etc.) compared to those with asthma or COPD alone, special referral for additional investigation is recommended in certain cases: persistent symptoms, severe and/or frequent exacerbations, diagnostic uncertainty and presence of comorbidities [78].

There are conflicting data on the prognosis of patients with ACO compared to COPD and asthma patients alone. In general, patients with ACO have been reported to experience more frequent and severe exacerbations, higher risk of hospitalization, accelerated decline in lung function, and poorer quality of life compared to patients with asthma or COPD alone [79,80,81,82,83]. However, data regarding the strongest outcome named mortality have been inconsistent [84]. There are studies that reported comparable mortality rates between ACO and COPD patients [84,85]. In contrast, some studies reported higher mortality rates in ACO subjects, while other studies reported lower mortality rates in patients with ACO compared to COPD alone [86,87].

Further studies are needed to clarify and establish a holistic approach to medical intervention in ACO patients and to investigate the accurate prognosis of ACO patients compared to asthma or COPD alone.

## 5. Future Directions

As diagnostic models are updated and improved, other diagnostic testing, such as the use of blood eosinophil or FeNO levels, has been an active area of investigation, yet has not shown benefits in distinguishing between phenotypes of chronic obstructive airway disease [88]. These findings highlight both the heterogeneity of the disease and the difficulty of identifying a singular manifestation of ACO.

As there is still ongoing debate over whether ACO is a discrete clinical entity or if it is part of a continuum of airways disease, long-term both observational and prospective studies are required to validate the various approaches to ACO diagnosis and to evaluate the role of currently available diagnostic tests. Moreover, genome-wide association studies may provide more comprehensive information regarding genetic, clinical, and imaging components associated with ACO. We also need to understand the underlying mechanisms of ACO, in order to identify specific biomarkers for diagnosis and targeted therapies and from this point of view, further validation studies and fundamental research are necessary to understand the link between genotypes and phenotypes. Likewise, given that abnormal lung growth affects fixed airway disease development later in life, prospective cohort studies from early adulthood would be necessary to evaluate the natural course and the prognosis of the disease. Finally, if we are to improve clinical outcomes for this complex and heterogeneous patient population, randomized controlled trials should be conducted on well-characterized patients with ACOS, especially in the elderly.

## 6. Conclusions

ACO historically includes patients with clinical features of both asthma and COPD; however, whether this term does define a single disease or is used descriptively to include various overlapping clinical phenotypes of chronic airways disease remains yet to be fully elicited. Nevertheless, despite the lack of a uniform ACO definition, emerging evidence supports the role of environmental exposures in ACO pathogenesis, and biomarker profiling and genetic analyses suggest that early-life factors and cigarette smoking may interact to increase the risk of airflow obstruction in certain susceptible subjects later in life.

In our opinion, this subgroup of patients deserves identification as a distinct clinical phenotype, as the observed differences in clinical presentation and treatment requirements, compared to patients with asthma or COPD alone, may impose a substantial burden in their clinical course and prognosis if they do not receive appropriate attention.

Nevertheless, the diagnosis of ACO can be difficult in everyday practice, due to the lack of a specific definition to differentiate it from asthma and COPD. Therefore, current evidence-based treatment recommendations should incorporate strategies that focus in defining specific and measurable treatable traits for every single patient. Advanced therapies in patients with ACO are based on extrapolated data from asthma and COPD and evidence-based treatment options for ACO based on defining biomarkers are a research priority. The understanding of the underlying etiological concepts that are responsible for the inflammatory and airway remodeling changes in ACO would probably improve the management and prognosis of these patients.

## Figures and Tables

**Figure 1 jpm-12-00708-f001:**
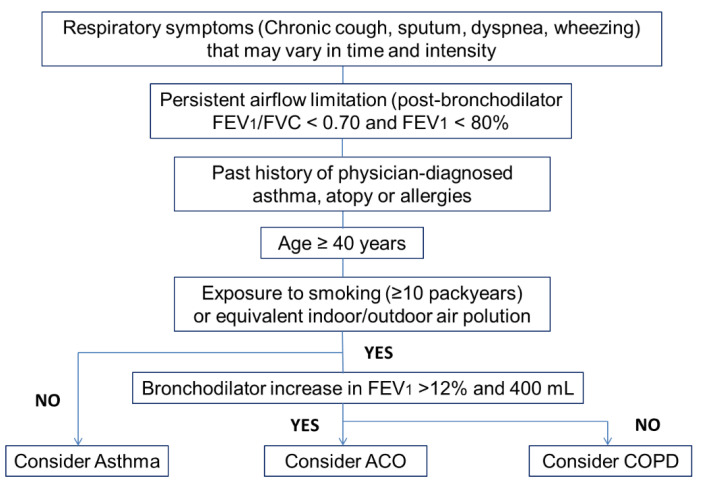
Flow-chart for diagnosing asthma-COPD overlap. ACO: asthma COPD overlap; COPD: chronic obstructive pulmonary disease; FEV1: forced expiratory volume in one second; FVC: forced vital capacity.

**Table 1 jpm-12-00708-t001:** Proposed diagnostic features of asthma and COPD overlap (ACO).

Study	Major Criteria	Minor Criteria	Diagnosis
Gibson, 2009 [3]	▪ Clinical symptoms of chronic airway disease, FEV_1_/FVC < 70% ▪ FEV_1_ <80% ▪ Bronchial hyper-responsiveness defined as a PD15 < 12 mL (provocative dose of hypertonic saline that induces a 15% fall in FEV_1_)		3 major criteria
Soler-Cataluna, 2011 [11]	COPD plus: ▪ Positive bronchodilator test defined by increase in FEV_1_ ≥ 15% and ≥400 mL ▪ Sputum eosinophilia ▪ History of asthma	COPD plus: ▪ High total serum IgE ▪ History of atopy ▪ Positive bronchodilator test, i.e., increase in FEV_1_ ≥ 12% and ≥200 mL over baseline on ≥2 occasions	2 major criteria OR 1 major criterion AND 2 minor criteria
Koblizek, 2013 [12]	COPD plus: ▪ Positive bronchodilator test defined by increase in FEV_1_ > 15% and >400 mL ▪ Methacholine challenge test positivity ▪ FE_NO_ ≥ 45 to 50 ppb and/or sputum eosinophils > 3% ▪ History of asthma	COPD plus: ▪ Mildly positive bronchodilator test, i.e., increase in FEV_1_ >12% and >200 mL ▪ Elevated IgE ▪ History of atopy	2 major criteria OR 1 major criterion AND 2 minor criteria
GINA/GOLD Criteria, 2014 [10]	More likely COPD if: ▪ Onset age > 40 years ▪ Persistence of symptoms ▪ Daily symptoms with exertional dyspnea and good/bad days ▪ Chronic cough and sputum precede onset of dyspnea, unrelated to triggers ▪ Documented persistent airflow limitation (post-bronchodilator FEV_1_/FVC <70%) ▪ Lung function abnormal between symptoms ▪ Previous physician diagnosis of COPD, chronic bronchitis or emphysema ▪ Heavy exposure to a risk factor (tobacco smoke, biomass fuel) ▪ Symptoms slowly worsening over time (progressive course over years) ▪ Rapid-acting bronchodilator treatment provides only limited relief ▪ Chest radiograph with features of severe hyperinflation	More likely asthma if: ▪ Onset age < 20 years ▪ Variation in symptoms within short periods ▪ Worsening of symptoms at night/early morning ▪ Symptoms triggered by exercise, emotions/laughter, dust, or allergens’ exposure ▪ Documented airflow limitation variability (peak flow, spirometry) ▪ Lung function normal between symptoms ▪ Prior physician diagnosis of asthma ▪ Family history of asthma or atopy/eczema ▪ No worsening of symptoms over time (symptoms vary either seasonally or from year to year) ▪ May improve spontaneously or have an immediate response to bronchodilators or to inhaled steroids over weeks ▪ Normal chest radiograph	If ≥3 items are present for either asthma or COPD, the patient is likely to have that disease A similar number of items for asthma and COPD is suggestive for ACO
Cosio, 2016 [13]	COPD plus: ▪ History of asthma ▪ Bronchodilator response to salbutamol > 15% and 400 mL	COPD plus: ▪ IgE > 100 IU ▪ History of atopy ▪ Two separated bronchodilator responses to salbutamol > 12% and 200 mL ▪ Blood eosinophils >5%	1 major criterion OR 2 minor criteria
Sin, 2016 [14]	COPD plus: ▪ FEV_1_/FVC < 0.7 or LLN in patients ≥ 40 years of age ▪ ≥10 pack years of tobacco smoking OR equivalent indoor or outdoor air pollution exposure ▪ Documented history of asthma before 40 years of age OR bronchodilator reversibility >400 mL in FEV_1_	COPD plus: ▪ Documented history of atopy or allergic rhinitis ▪ Bronchodilator reversibility of FEV_1_ ≥ 200 mL and 12% from baseline on ≥2 visits ▪ Peripheral blood eosinophil count of ≥300 cells/mL	3 major criteria AND 1 minor criterion
Cataldo, 2017 [15]	ACO in a COPD patient: ▪ High degree of variability in airway obstruction over time: FEV_1_ variation ≥ 400 mL ▪ High degree of response to bronchodilators: >200 mL and 12% above baseline	ACO in a COPD patient: ▪ Personal or family history of atopy and/or IgE sensitivity to one or more airborne allergens ▪ Elevated blood or sputum eosinophils or increased FE_NO_ ▪ Asthma diagnosed before the age of 40 ▪ Symptoms’ variability ▪ Age (in favor of asthma)	2 major criteria AND 1 minor criterion
	ACO in an asthma patient: ▪ Persistence over time of airflow obstruction (FEV_1_/FVC <0.7 or <LLN) ▪ Exposure to noxious particles or gases, with ≥10 pack years for smokers	ACO in an asthma patient: ▪ Lack of response on acute bronchodilator tests ▪ Reduced lung diffusion capacity ▪ Little variability in airway obstruction ▪ Age in favor of COPD (>40 years) ▪ Presence of emphysema on chest CT scan	
Miravittles, 2017 [16]	▪ Age > 35 years ▪ Postbronchodilator FEV_1_/FVC < 70% ▪ ≥10 pack years tobacco smoke	▪ Current diagnosis of asthma ▪ No current diagnosis of asthma but a bronchodilator response to albuterol ≥ 15% and 400 mL and/or blood eosinophils ≥ 300 cells/microL	3 major criteria AND 1 minor criterion

ACO: asthma COPD overlap; COPD: chronic obstructive pulmonary disease; IgE: immunoglobulin E; FE_NO_: fraction of exhaled nitric oxide; ppb: parts per billion; FEV_1_: forced expiratory volume in one second; FVC: forced vital capacity; IU: international units; LLN: lower limit of normal; PD15: provocative dose.

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
