# Peer review of "Asthma-COPD Overlap Syndrome: Recent Insights and Unanswered Questions"

_jpm, 2022, doi:10.3390/jpm12050708_

Round 1

Reviewer 1 Report

Overall the review is well presented and succinct and makes for an easy read and is very relevant to clinical practice 

I would like the authors to incorporate a section where in they further discuss a hypothesized /proposed schema for investigating and diagnosing ACO. I would also like the authors to incorporate a section which discusses the challenges in diagnosing ACO and future directions. 

Reviewer 2 Report

The paper tries to summarize different aspects of the ACO. Considering the whole manuscript,  the authors explore different aspects of this clinical entities but there are some features that need to be improved.  

Since ACO has been rejected by two of the major society (namely GOLD and GINA) in 2021, as reported in the description paragraph, the controversy about this existence should be illustrated better and also in the introduction and in the conclusions the authors should explain  the reason for believing in ACO or not. 

In the second section, the definition, it should be better described the clinical features that are correlated to ACO even the minor criteria and it should be put a reference to the table that reassume all the studies on the topic.

The pathogeny paragraph is lacking in any characterization of the two kind of patients that could fulfill the ACO criteria (namely COPD with increased reversibility and asthma with fixed airflow obstruction) from the inflammatory point of view (ex. eosinophils vs neutrophils). This explains  the need of better phenotyping the obstructive patients observed in the past years and the ACO idea itself.

In the genetic paragraph the word "indicate" should be modified. In fact  the studies described as in favor to demonstrate a genetic basis did not reach the significance even if they identify some SNPs that could be associated with ACO. 

The reference 12 seems to be wrong: should it be  a reference for a GINA report?

Round 2

Reviewer 2 Report

There is a phrase “ACO Challanges” that I don’t understand just before paragraph 4. Is it a new paragraph or just a typing error?

Author Response

I would like to thank the reviewer for his/her comment. It is actually the title of a new subsection of section 3 with number 3.9 and title "3.9. Challenges in ACO diagnosis".

This subsection has been added according to reviewer's #1 comments and its numbering has been ommited by mistake.